# Human iPSC-Cardiomyocytes as an Experimental Model to Study Epigenetic Modifiers of Electrophysiology

**DOI:** 10.3390/cells11020200

**Published:** 2022-01-07

**Authors:** Maria R. Pozo, Gantt W. Meredith, Emilia Entcheva

**Affiliations:** Department of Biomedical Engineering, George Washington University, Washington, DC 20052, USA; mpozo@gwu.edu (M.R.P.); meredithg@gwu.edu (G.W.M.)

**Keywords:** human iPSC-CMs, cardiac electrophysiology, cardiac epigenetics, histone deacetylases, histone acetyltransferases, HDAC inhibitors, DNA acetylation

## Abstract

The epigenetic landscape and the responses to pharmacological epigenetic regulators in each human are unique. Classes of epigenetic writers and erasers, such as histone acetyltransferases, HATs, and histone deacetylases, HDACs, control DNA acetylation/deacetylation and chromatin accessibility, thus exerting transcriptional control in a tissue- and person-specific manner. Rapid development of novel pharmacological agents in clinical testing—HDAC inhibitors (HDACi)—targets these master regulators as common means of therapeutic intervention in cancer and immune diseases. The action of these epigenetic modulators is much less explored for cardiac tissue, yet all new drugs need to be tested for cardiotoxicity. To advance our understanding of chromatin regulation in the heart, and specifically how modulation of DNA acetylation state may affect functional electrophysiological responses, human-induced pluripotent stem-cell-derived cardiomyocyte (hiPSC-CM) technology can be leveraged as a scalable, high-throughput platform with ability to provide patient-specific insights. This review covers relevant background on the known roles of HATs and HDACs in the heart, the current state of HDACi development, applications, and any adverse cardiac events; it also summarizes relevant differential gene expression data for the adult human heart vs. hiPSC-CMs along with initial transcriptional and functional results from using this new experimental platform to yield insights on epigenetic control of the heart. We focus on the multitude of methodologies and workflows needed to quantify responses to HDACis in hiPSC-CMs. This overview can help highlight the power and the limitations of hiPSC-CMs as a scalable experimental model in capturing epigenetic responses relevant to the human heart.

## 1. Introduction

Epigenetic studies offer insights into the modulation of human gene expression by environmental stimuli [1]. Organ specificity [2] and the dynamic nature of epigenetic regulation over space and time, driven by a variety of environmental factors [3], can greatly impact cardiac function. Advancements in induced pluripotent stem cell technology [4,5] have yielded a valuable in vitro model of the human heart, human-induced pluripotent stem cell cardiomyocytes (hiPSC-CMs), which may offer a platform for such cardiac epigenetic studies. It is important to understand if this in vitro model of the human heart can recapitulate the in vivo complexity. Here, we review preliminary studies of this model and discuss aspects related to the use of hiPSC-CMs to gain insights into epigenetic regulation of cardiac electrophysiology, specifically as related to the function of histone deacetylases.

### 1.1. hiPSC-CMs as a Scalable Model of Cardiac Electrophysiology

hiPSC-CMs, originally derived from human fibroblasts [4], and more recently from noninvasive sources such as blood [6], offer patient-specific cardiomyocytes [7] for a range of applications. Over the past 15 years, induced pluripotent stem-cell technology has undergone active development and optimization towards a more mature phenotype [5], impacting the fields of disease modeling, personalized therapeutics, tissue engineering, regenerative medicine, and drug cardiotoxicity screening. For example, hiPSC-CMs have been used in cardio-oncology applications [8], where replicable cell sources are necessary for large-scale genetic screenings [9]. hiPSC-CMs offer said scalability, derived from a seemingly limitless stem-cell source useful in cardiac pathophysiological studies [10], as evidenced in long QT syndrome [11] and Torsades de Pointes (TdP) [12], among others.

The high-throughput potential and scalability offered by hiPSC-CMs are particularly attractive in preclinical cardiotoxicity screening; hiPSC-CMs are used to perform “clinical trials in a dish” [13], and have been shown to recapitulate aspects of human clinical data in vitro [14]. A crucial benefit of hiPSC-CM “clinical trials” over the traditional clinical trial is the ability to perform well-powered studies under controlled conditions to gain mechanistic insights. In vitro all-optical electrophysiology [15,16] can be leveraged to obtain a comprehensive interrogation of the electromechanics in hiPSC-CMs. Long-term studies of hiPSC-CMs in a high-throughput setting have also been demonstrated recently [17,18,19,20,21,22], supporting the feasibility of a human experimental model for chronic functional cardiac studies. Derivation and full characterization of new patient-derived iPS lines is still a tedious and time-consuming process and is often centralized into several key groups that contribute to databases of healthy and diseased iPS lines. Furthermore, in-house differentiation can yield variable results even when similar protocols are followed. However, for preclinical testing applications, commercially-available and reproducible predifferentiated human iPSC-CM lines are typically used, which allows direct comparison of results from different testing facilities.

### 1.2. Human Experimental Models Are Needed for Functional Cardiac Studies

While animal models, especially transgenic mice [23], have been the norm for physiological and drug screening studies, issues such as interspecies epigenomic variation, cell-specific ion channel response variation, and renal clearance variation [24] present significant limitations in clinical translation. For example, rodents can exhibit species-specific ion channel responses [25], such as key cardiac potassium currents [26], and therefore are not ideal for studying human cardiac electrophysiology. Fundamental measurements, such as ECG, can differ greatly from species to species [27]. Comparison of human and animal models, such as mouse and pig, revealed high epigenomic variability between species [28,29] and even within humans, sex-specific cardiac differences exist, including heart size, hormone–cardiac–electrophysiology interplay, and response to environmental (temperature) stimuli [30]. Such differences, present in native physiology, further emphasize the need for a patient-specific, human model that is both reliable and translatable.

### 1.3. hiPSC-CMs for Drug Cardiotoxicity Screening

Detection of drug cardiotoxicity presents a significant challenge for both pharmaceutical development and clinical decision-making regarding treatment. Cardiotoxicity can manifest in many forms, such as ECG abnormalities, biochemical markers such as natriuretic peptides, and blood pressure variation [31]. In vitro, observation and detection of cardiotoxicity is enhanced by scalable and comprehensive analysis of cellular responses [15,16], i.e., noninvasive measurements of key electrophysiological parameters, such as action potential duration (APD), calcium handling properties, and contractile properties that can be used to predict and model drug response. Currently, there are two major avenues of assessing drug response in patients: animal studies and human clinical trials. Current preclinical drug cardiac studies (Figure 1A) are often performed in dogs, pigs, or mice [23], which, as described previously, presents significant limitations. Further, in vivo animal models require lengthy and costly protocols [32], and are therefore not scalable. The current human cardiotoxicity model (Figure 1B) is majorly limited by patient availability, time, and consistency. Clinical trials can last many years and only about eight percent of trials progress to Phase IV [33]. Due to patient-inaccessibility, trial length, and cost, hiPSC-CMs have been proposed as an alternative model for cardiotoxicity screening (Figure 1C). hiPSC-CMs have proven successful in recapitulating patient-specific disease phenotypes [34,35,36,37] and drug-induced cardiotoxicity [38]. hiPSC-CMs offer the ability to observe phenotypical, epigenetic, and functional responses in hiPSC-CMs for improved cardiotoxicity studies.

hiPSC-CMs can be obtained in far greater abundance [39] than other animal or human samples. Functional measurements in hiPSC-CMs, including optical mapping, predict in vivo effects without the need for invasive cardiac sample retrieval [40,41]. However, in vitro models are inherently limited by oversimplified microenvironments. Novel cell culture systems, such as coculture of hiPSC-CMs and cardiac fibroblasts [42] or other multicellular assemblies and 3D bioprinting [16,43,44], address this limitation through introduction of physiologically-relevant cell lines, reagents, and mechanical conditions (e.g., microfluidics [22]). Specifically, three-dimensional cells for preclinical testing can be obtained through cell-seeded hydrogel molding of engineered heart tissues [43], light-based (DLP-driven) bioprinting of cell-seeded hydrogels [45], and scaffold-free modular approaches, such as cardiac spheroids assembled on needle arrays in designer macrostructures [44], among others. The modularity, scalability, and drug/disease screening potential of hiPSC-CMs is well-supported [12,18,41,43,46,47,48,49]. Thus, hiPSC-CMs have emerged as a promising model for studying cardiac electrophysiology, and this review discusses the potential application of this model to epigenetics studies of the human heart.

## 2. Epigenetic Modulators of the Cardiovascular System

Recently, cardiac studies have implied broader interests in epigenetics. Epigenetic modulators are master regulators of fundamental cellular processes such as cell development, cell survival, and cell death. In the heart, this regulation is evidenced in cell fibrosis, hypertrophy, and ischemia/reperfusion injury, to name a few. The general role of epigenetic modulators in these processes is well understood; however, their specific impact on cell electrophysiology is not studied extensively. In cardiac-related studies, electrophysiological parameters become increasingly relevant, as they are among the key determinants of drug withdrawal from market. We discuss here the epigenetic landscape of the human heart, focusing on histone deacetylases (HDAC) and their role in hiPSC-CM electrophysiology.

### 2.1. Epigenetic Regulators in the Heart and Control of Cardiac Electrophysiology

Epigenetic modulation of chromatin structure and gene expression is performed by “writers, erasers, and readers”—enzymes which add, remove, and interpret post-translational modifications (PTMs; acetylation, methylation, phosphorylation, ubiquitination) on histone proteins, typically targeting amino acid residues in the histones’ N-terminal tail, to mediate action on DNA transcription. Reader domains have a high affinity for PTM sites and can be found in numerous proteins, including chromatin modifying proteins and chromatin remodeler or adaptor proteins. In the heart, certain reader domains are particularly critical. For example, the bromodomain and extraterminal domain (BET) family of adaptor proteins contains bromodomains that recognize sites of histone acetylation and interact with transcription machinery [50]. Brg1, a BET protein, associates with chromatin remodeling complexes such as SWI/SNF, which is implicated in the regulation of a variety of genes [51,52]. This association between Brg1 and SWI/SNF is a key component for reading histone acetylation in the human heart [50].

Writers and erasers cooperatively mediate the PTM addition–removal axis. Among important writer–eraser pairs in the human heart are histone methyltransferases (HMTs) and histone demethylases (HDMs), which methylate and demethylate histone tails, respectively. Certain HMTs (such as SMYD1, WHSC1, Ezh2, SUB39h, and DOT1L) and HDMs (Jmjd1–Jmjd3) contribute to cardiac development and hypertrophy [53,54,55]. Control of transcription by these enzymes can be either repressive or enhancing, depending on the location and extent of methylation (reviewed in several sources [50,56,57]). In addition to histone methylation, histone acetylation is another critical PTM and is controlled by histone acetyltransferases (HATs) and histone deacetylases (HDACs). HATs and HDACs are considered master regulators of gene transcription due to their role in chromatin remodeling, where histone acetylation loosens chromatin structure and histone deacetylation “winds” chromatin into a tighter structure (Figure 2) inaccessible to transcription machinery. HAT enzymes p300 and pCAF are particularly important in the human heart, contributing to cardiac development and hypertrophy [50,58,59,60,61]. HDACs are unique in their ability to deacetylate not just histones but a broad range of protein targets, affecting diverse cellular processes including chromatin remodeling, cell cycle, splicing, and microtubule stabilization [62]. Four main classes of human HDACs exist: class I (HDAC1, HDAC2, HDAC3, HDAC8), class II (IIa: HDAC4, HDAC5, HDAC7, HDAC9; IIb: HDAC6, HDAC10), class III (SIRT1-7), and class IV (HDAC11) (Table 1). Classes I, II, and IV are classical, Zn^2+^-dependent HDACs, while class III HDACs are NAD^+^-dependent.

While HDACs and their roles in fundamental cell processes such as development, cell life, and cell death are generally well studied, their role in cardiac electrophysiology is not fully characterized. HDAC activity influences cardiac electrophysiology through two general mechanisms. First, HDACs perform transcriptional reprogramming. HDAC classes I and II coregulate fetal gene programs in cardiomyocytes [86,87] which, during cardiac stress, result in impaired myocyte contractility [88,89]. Class I HDACs also act on important electrophysiology-modifying TFs such as NF-κB, which is involved in Ca^2+^ handling, and NKX2.5, which is a critical TF for the Na^+^/Ca^2+^ exchanger (*NCX1*) [90,91]. HDACs 1, 3, and 5 form a corepressor complex that deacetylates NKX2.5, promoting *NCX1* expression [91,92]. HDAC4 inhibits myocyte enhancer factor (MEF2) and suppresses Ca^2+^ pathways in the heart [93].

The second mechanism for HDAC impact on cardiac electrophysiology is through interactions with cytoskeletal and contractile proteins in the heart [62,94,95]. For example, α-tubulin, which is important for gap junction growth and connexin trafficking [96], is critical for cardiac biomechanics [97] and is a target of HDAC6, a predominantly cytoplasmic deacetylase [98,99]. HDAC6 action on α-tubulin leads to degradation and, ultimately, structural and contractile dysfunction [100]. In addition, an important cytoskeletal HDAC6 target [101,102] is cortactin, which associates with K_V_1.5 to regulate I_Kur_ [103]. Deacetylation of cortactin by HDAC6 leads to loss of Ca^2+^ transients and can cause arrhythmogenesis through perturbation of K^+^ currents [103]. Other HDACs can interact with contractile proteins by binding directly to myofibrils [104]. For example, HDAC4 deacetylates muscle LIM protein, which reduces myofilament Ca^2+^ sensitivity [94]. Table 1 summarizes the general cardiac involvement of each HDAC, highlighting potential heart-healthy or detrimental roles.

The HDAC/HAT axis is crucial for cardiac hypertrophy and development. Because class I HDACs are known to promote growth in the heart [65,67,105,106,107,108,109,110,111] while class II HDACs generally suppress growth and fetal gene activation [57,73,112,113], class I HDACs are broadly regarded as pathological in the heart while class II HDACs are regarded as cardioprotective [114]. However, key studies in this field are performed in animal models. Therefore, careful evaluation in humans is necessary. Of particular interest are effects of HDAC/HAT on essential cardiac channels (Figure 2) such as the Na^+^/Ca^2+^ exchanger, the Kir2.1 ion channel, and the Na^+^/K^+^ pump [115], revealing underlying mechanisms of epigenetic regulation of cell electrophysiology.

### 2.2. Differential Gene Expression in hiPSC-CMs and the Adult Heart Relevant to Epigenetic Modifiers and Electrophysiological Function

Evaluation of hiPSC-CMs as an experimental tool in epigenetics requires consideration of the cells’ epigenetic profile and how it compares to that of the adult heart. Few studies have robustly investigated this comparison, hence only a small pool of literature exists where expression profiles of epigenetics genes were examined in both hiPSC-CMs and adult heart. Two early studies performed such comparative microarray experiments [116,117], which are discussed here.

Babiarz and colleagues analyzed in-house-derived hiPSC-CM RNA and adult human cardiac RNA (Ambion, Austin, TX, USA) using Illumina HumanWG6-V3 BeadChip (Illumina, San Diego, CA, USA), while the Gupta group processed in-house-derived hiPSC-CM RNA and adult human heart RNA (Clontech, Saint-Germain-en-Laye, France) using Sentrix^®^ Human HT-12_V3 whole-genome bead chips (Illumina, San Diego, CA, USA).

Expression data for cardiac-relevant epigenetics genes [50] showed general upregulation of HDAC, HMT, HDM, and reader genes in hiPSC-CMs compared to adult heart (Table 2A). These differences are not extreme overall (fold change within ±2) but are of note. Key cardiac electrophysiology genes, however, presented more drastic observations (Table 2B). For example, *KCNJ2*, a gene coding for the Kir2.1 channel which contributes to the inward-rectifying potassium current I_K1_, is substantially downregulated in hiPSC-CMs compared to adult heart—a known deficiency and a common target for optimization of maturity in hiPSC-CMs [118], for example through genetic overexpression of *KCNJ2* [119]. Differential expression of cardiac electrophysiology genes yields no clear trend, indicating the need for further robust quantification. Moreover, microarray technology as well as protocols for hiPSC-CM differentiation and maturation have seen substantial improvements in recent years [119,120,121]. Three-dimensional approaches [43,44,45] to create more physiologically-relevant cell structures may further affect maturity and the potential of hiPSC-CMs to better capture epigenetic modulations. Therefore, more up-to-date comparative transcriptomics experiments are needed. This work may further benefit from RNA interference (RNAi) studies where perturbation of specific epigenetic genes (such as those in Table 2A) would reveal impacts on expression of cardiac electrophysiology genes.

## 3. Pharmacological HDAC Inhibition

HDAC inhibitors (HDACi), a unique class of epigenetic modifiers, have gained prominence in the laboratory and as therapeutic agents (Table 3) for a variety of pathologies, especially cancer [122,123,124] and autoimmune diseases [125]. Over the past two decades, four FDA-approved HDACi have emerged: vorinostat (SAHA) [126], romidepsin (depsipeptide) [127], belinostat (PXD-101) [128], and panobinostat (LBH589) [129]. Data from clinical trials, animal models, and in vitro studies have revealed both negative and positive effects of HDACi on heart health.

### 3.1. HDACi in Clinical Trials and Post-Market Observations

Pharmacological inhibition of HDACs is an active field of research, with 194 ongoing (Figure 3A) and 373 successfully completed trials in the US to date. Vorinostat, panobinostat, and entinostat predominate these studies. Cancer indications are most prevalent, contributing over 90% (180) of all ongoing HDACi clinical trials, but other indications, such as HIV (7), are also studied (Figure 3B). Data from these FDA trials suggests limited cardiotoxicity with reports of ST/T abnormalities, QTc prolongation, heart failure, and hypotension, among others.

Schiattarella and colleagues analyzed results from sixty-two HDACi clinical trials and interestingly found that although panobinostat and romidepsin had high rates of cardiotoxicity (~30% incidence), the severity of observed effects was quite low, with over 70% of events being grade 0–1 [130]. Conversely, vorinostat and belinostat had low rates of cardiotoxicity (<15% incidence) but were more severe, with over 50% of observed cardiac side effects being grade 3–4. Results from the FDA Adverse Events Reporting System (FAERS) also report HDACi cardiotoxicity in the form of atrial fibrillation, TdP, tachycardia, cardiac failure, and many others (Figure 4). In general, high toxicity is associated with pan-HDACi (vorinostat, panobinostat) and progress towards isoform-selective HDACi aims to improve safety.

Compared to other anticancer therapies, HDACi show generally little cardiotoxicity [131]. Of note, however, is “hidden cardiotoxicity” [132], which manifests only in the diseased heart. Moreover, HDACi have been clinically associated with delayed cardiotoxicity, where onset of adverse effects such as atrial fibrillation and ventricular tachycardia occurs several hours up to days after treatment [133]. This delayed onset impedes early detection in drug safety testing, which is typically focused on acute, rather than delayed, side effects. Because many safety studies are performed in heart-healthy individuals, and because HDACi generally have minimal direct acute effects (≤1 h), cardiotoxicity screening for HDACi requires an improved platform that can address hidden cardiotoxicity through chronic monitoring of both healthy and diseased models.

**Table 3 cells-11-00200-t003:** Specificity and application of pan and selective HDACi. “+” indicates inhibitory selectivity. Additional “+” and red color indicate greater inhibitory effect. Modified from [134].

	Class I				Class IIa				Class IIb		Class IV		
**Inhibitor Name**	**HDAC1**	**HDAC2**	**HDAC3**	**HDAC8**	**HDAC4**	**HDAC5**	**HDAC7**	**HDAC9**	**HDAC6**	**HDAC10**	**HDAC11**		Clinicaltrials.gov (02/22/2019)
Vorinostat (SAHA)	++++	++++	++++	++++	++++	++++	++++	++++	++++	++++	++++	Merck (FDA)	251
Panobinostat	++++	++++	++++	++++	++++	++++	++++	++++	++++	++++	++++	Novartis (FDA)	133
Trichostatin A	++++	++++	++++	++++	++++	++++	++++	++++	++++	++++	++++		15
Belinostat	++++	++++	++++	++++	++++	++++	++++	++++	++++	++++	++++	TopoTarget (FDA)	44
Dacinostat	++++	++++	++++	++++	++++	++++	++++	++++	++++	++++	++++	Novartis	-
M344	++++	++++	++++	++++	++++	++++	++++	++++	++++	++++	++++		-
AR-42	++++	++++	++++	++++	++++	++++	++++	++++	++++	++++	++++	Arno Therapeutics	5
Quisinostat	++++	++++	++++	++++	++++	++++	++	+++	++	++++	++++		6
CUDC-907	++++	++++	++++	++	++	+	++	++	+++	++++	++++		6
Pracinostat	+++	++	+++	++	+++	+++	++	++	+	+++	++	MEI Pharma	12
CUDC-101	++++	+++	++++	++	+++	+++	++	++	++++	+++		Curis	4
Ricolinostat	++	+++	+++	++	+	+	+		++++			Celgene/Acetylon	9
Abexinostat	++++	+++	++++	++					+++	+++		Pharmacyclics	9
HPOB	+	+	+	+					+++	+			1
MC1568	++	++	++	++									-
Mocetinostat	++	++	+								+	Mirati	22
TMP269					++	++	+++	+++					-
PCI-34051	+			++++					+	+			-
Droxinostat			+	+					+				-
Resminostat	+++		+++						++			4SC	5
BRD72954		+		++					+++				-
BG45	+	+	++										-
4SC-202	+	+	+									4SC	3
Tacedinaline	+	+	+										3
LMK-235					+++	++++							-
Romidepsin	+++	+++										Celgene (FDA)	88
RG2833	+++											Replign	-
Entinostat	++		+									Syndax	60
CAY10603	++								++++				-
Tubacin									++++				-
RGFP966			++										-
Tubastatin A									+++				-
Nexturastat A									++++				-
SS-2-08									++++				-

### 3.2. HDACi Have Cardiac Therapeutic Potential

While the focus of HDACi therapy has been centered on cancer and autoimmune disease, HDACi have also exhibited cardiac therapeutic potential in animal models and in vitro studies [135] (reviewed in [136]). Treatment with Trichostatin A (TSA) suppressed pathological cardiac hypertrophy in transgenic mice [108,137]. In other studies, TSA and pan-HDACi scriptaid blunted cardiac hypertrophy and enhanced ventricular performance, improvements that were maintained over nine weeks of treatment in a pressure-overload mouse model, demonstrated long-term efficacy of TSA [107]. Pan-HDACi have also reduced maladaptive ventricular remodeling and improved cardiac performance in rodent models of myocardial infarction [70,138,139] and in chronic hypertension rat models [140,141]. Cao and colleagues found that excessive cardiac autophagy, which contributes to the pathogenesis of heart failure [142], was blocked by TSA in a pressure-overload murine model [110]. They also used RNAi to demonstrate a role for HDACs 1 and 2 in agonist-dependent cardiac autophagy. Work by Wallner and colleagues revealed that vorinostat reduced left ventricular hypertrophy, left ventricular diastolic dysfunction, and atrial remodeling in a feline diastolic dysfunction model [143]. Overall, the benefits of HDACi in cardiac applications stem from their broader role as chromatin modifiers, controlling key cardiac transcription factors, assisting in chaperone activity and protein quality control, as well as serving as mediators of cell signaling and metabolic health [135,136]. Observations of both cardiac therapeutic potential and cardiotoxicity of HDACi in the heart necessitate a robust platform for quantification and characterization of cardiac HDACi effects.

## 4. Methodologies for Quantifying Effects of HDACi in hiPSC-CMs

The hiPSC-CM model for high-throughput screening of cardiotoxicity is made possible by a robust testing “toolbox” through which genetic, epigenetic, and phenotypical analysis can be performed in a quantitative manner. Described here are, following experimental workflow, the techniques used for quantification of HDACi effects on hiPSC-CMs. We consider an experimental setup including a treatment group (HDACi treatment: various concentrations, various HDACi drugs) and a control group of untreated hiPSC-CMs. Of interest in this scheme is quantifying various stages of HDACi effect, beginning with inhibition of HDAC activity and continuing through to changes in cell functional behavior (Figure 5).

### 4.1. Quantification of HDAC Enzyme Inhibition

After application of HDACi drug to cultured hiPSC-CMs, HDAC activity assay kits, such as HDAC-Glo™ I/II Assay (Promega, Madison, WI, USA), quantify HDAC enzymatic activity. Figure 5(Aii) illustrates the assay mechanism, where a proprietary acetylated substrate is deacetylated by the HDACs in the cell sample. Availability of the deacetylated substrate is measured through aminoluciferase-based chemiluminescence, recorded by a microplate photometer (i.e., plate reader). Reduced chemiluminescence is expected for HDACi-treated samples. Because HDAC-Glo™ I/II is performed directly on cultured cells in a single-reagent-addition format and because the luminescent signal has a half-life of >3 h, this assay is amenable to high-throughput and/or automated experimental platforms.

### 4.2. Quantification of Histone Acetylation: ATAC-seq, ChIP-seq, Western Blot

HDAC inhibition corresponds to loose chromatin structure and high levels of chromatin acetylation (Figure 5(Bi)). There are several options for quantification (extensively reviewed in [145]). The assay for transposase-accessible chromatin using sequencing (ATAC-seq) (Figure 5(Bii)) quantifies loci-specific chromatin accessibility [146]. Tn5 transposase binds open chromatin and inserts adaptor sequences. The resulting fragments are deep-sequenced to map regions of open chromatin. Increased chromatin accessibility is expected in HDACi-treated samples, and loci-specific data may shed light on epigenetic mechanisms of HDACi effects on cardiophysiology. Library preparation for ATAC-seq requires roughly 50,000 cells [147] and the ATAC-seq workflow was designed for sequencing using high-throughput instruments such as Illumina, making this chromatin accessibility assay accordant with high-throughput experimentation in hiPSC-CMs.

Chromatin immunoprecipitation sequencing (ChIP-seq) is another method for identifying open regions of chromatin (Figure 5(Biii)) [148,149,150]. ChIP-seq reveals localization of chromatin-binding proteins such as transcription factors (TF) or other transcription machinery. In brief, chromatin is sheared [145] and antibody pulldown isolates the protein of interest and its bound DNA is collected for deep sequencing or quantitative polymerase chain reaction (qPCR). ChIP-seq can be applied to cardiac epigenetics studies to reveal chromatin accessibility at loci associated with critical electrophysiology TF (Table 1). Combination of chromatin immunoprecipitation with high-throughput sequencing technology in ChIP-seq is compatible with small-volume, precious sample material, such as hiPSC-CMs.

Histone acetylation levels are also affected by HDAC inhibition and can be quantified using western blot, WB (Figure 5(Biv)) [151,152,153]. While standard WB remains prevalent, this method is not suitable when using expensive samples due to large input requirement, low throughput, and lengthy protocol. Recent WB miniaturization using capillary electrophoresis addresses such concerns to allow handling of smaller samples, compatible with 96-well plates [154].

A recently developed analysis strategy involves overlay of maps from Genome-Wide Association Study (GWAS), ATAC-seq, and ChIP-seq (Figure 5(Bv)) [155]. By aligning these maps, loci can be identified where chromatin accessibility, TF of interest, and certain phenotypic traits (GWAS) are all enhanced. In cardiac epigenetics studies, this technique may provide unmatched insights into underlying links between HDACs, their target TFs, and resulting cardiac physiological phenotypes, known from GWAS.

### 4.3. Quantification of Gene Transcription: qPCR, Microarray, RNA-seq

Because chromatin accessibility impacts gene expression, RNA profiling reveals output of HDACi epigenetic effects (Figure 5(Ci)). Quantification of gene expression can be performed in multiple ways. qPCR (Figure 5(Cii)) is a well-established technique where isolated RNA is reverse-transcribed and the complementary sequence (cDNA) is amplified using either probes (ex TaqMan) or nonspecific dyes (ex SYBR Green) [156,157]. While traditional qPCR is low-throughput and not suitable for small sample volumes, newly available Cells-to-C_T_ ™ (Invitrogen, Carlsbad, CA, USA) is compatible with 96-well format and is a considerable improvement to the qPCR workflow. Still, only a small number of transcripts can be probed simultaneously with qPCR, which is not intended for large-scale transcriptomics.

Prominent developments in gene expression quantification include microarray and RNA-seq, extending to spatial [158] and single-cell [159] transcriptomics (scRNA-seq). scRNA-seq provides unmatched preservation of complex transcriptomics relationships. However, this tool has limitations when used with large cells, such as hiPSC-CMs, where nuclear extraction is required. Other technologies, though, such as microarray, have been robustly used with hiPSC-CMs. In microarray profiling (Figure 5(Ciii)), extracted RNA is reverse-transcribed. The cDNA is fluorescently labeled and hybridized to microarrays against various genomic loci, with current microarrays offering coverage for hundreds of thousands of probes. With advancement of microarray technology, RNA input requirements have decreased drastically from ~1 ug per sample to as little as 100 ng per sample, improving throughput and feasibility for precious samples. In the context of HDACi effects on hiPSC-CMs, expression levels of key cardiac electrophysiology and cardiac epigenetics genes (Table 2) are particularly interesting.

The most powerful emerging high-throughput alternative to qPCR is RNA-seq, a next-generation sequencing-based platform (Figure 5(Civ)). Briefly, the prepared RNA library undergoes cyclic cross-bridge amplification, and unique fluorescent labeling of each nucleotide (A, T, C, G) allows nucleotide-by-nucleotide sequencing of every RNA fragment simultaneously. RNA-seq has progressed to surpass microarrays in usage and to require only small amounts of RNA—critical for precious samples. RNA-seq is distinct from microarray profiling in that RNA-seq allows full sequencing quantification of all RNA fragments and is, therefore, a deeper technology leveraged as a discovery tool, whereas microarray only allows profiling of known transcripts.

### 4.4. Quantitative Functional Studies

The translation of HDACi effects on gene expression into changes in functional behavior must be investigated. In addition to microelectrode arrays (MEAs) used for functional recordings [133], all-optical electrophysiology has emerged as a technique of choice [15,16,18,160,161]. The need to assess chronic HDACi effects necessitates the use of genetically-encoded voltage and/or calcium sensors. For example, R-GECO, jR-GECO, a red-shifted genetically-encoded calcium indicator [162,163], has been used in combination with optogenetic actuators such as ChR2, a genetically-encoded light-sensitive ion channel [164,165], for cardiac applications [15,22,160]. R-GECO can also be used with near-infrared (NIR) optogenetic voltage indicators such as Quasars [166] or high-performance NIR synthetic voltage probes such as BeRST [15,167] or di-4ANBDQBS [15,168]. Optogenetic sensors and actuators can be engineered into stable iPS cell lines, pre-differentiation, targeting the AAVS1 safe harbor site to avoid off-target effects, and to permit chronic continuous monitoring and control. For example, a dual-reporter cell line with a genetically-encoded calcium sensor, GCaMP6f, and a nuclear reporter, RedStar, was constructed recently [169]. Alternatively, plasmid transfection and adenoviral or lentiviral vectors provide straightforward means for expressing optogenetic actuators or sensors into differentiated and matured iPSC-CMs just before their deployment in drug screening assays. Such studies include the use of the Optopatch platform with spectrally-compatible optogenetic actuator CheRiff (a blue-shifted version of ChR2) and NIR quasar voltage sensor [161,166]. In a drug-testing application, Dempsey et al. combined hiPSC-CMs expressing CaViar, a dual voltage-calcium sensor system, with cells expressing CheRiff—an optogenetic actuator [160]. The benefit of optogenetic transformation post-differentiation or the usage of small-molecule dyes is the flexibility to use any disease model cell line or commercially-differentiated cells. Lentiviral expression of optogenetic tools is stable and allows longer-term monitoring, just as the reporter cell lines.

All-optical platforms, e.g., Optopatch [160,161,166], OptoDyCE [15,16], which integrates optogenetic actuators with optogenetics sensors and/or synthetic voltage probes (e.g., BeRST), allows for multiparametric assessment of function and response to pharmacological or genetic perturbations. The characterization includes simultaneous recordings of cell action potentials (AP), Ca^2+^ transients, and cell contraction (Figure 5D). Such platforms can be designed at low cost, are inherently scalable, high-throughput (automated with design for 96-well format), and all-optical electrophysiology has been robustly validated in hiPSC-CMs [15,16,18,22,170,171,172]. Unlike MEAs, they are compatible with 3D cell constructs as well. Cardiac all-optical electrophysiology can be leveraged in hiPSC-CM epigenetics studies to indicate and quantify cell responses to HDACi administration, where the multiparameter investigation can capture cardiotoxicity or cardioprotective HDACi effects.

## 5. Epigenetic Studies in hiPSC-CMs

Epigenetics-related studies in hiPSC-CMs are only just emerging, and key investigations are listed and discussed here (Table 4).

### 5.1. Epigenetic Characterization of hiPSC-CMs

Several studies have investigated the epigenomic profile of hiPSC-CMs, particularly through collection of ATAC-seq, ChIP-seq, and RNA-seq data (Table 4) [155,173,174]. Notably, Benaglio and colleagues produced large epigenetic datasets and cross-referenced ATAC-seq and ChIP-seq maps to gain novel insights into epigenetic mechanisms. Alignment of ATAC-seq and ChIP-seq maps reveals areas where accessible chromatin, histone acetylation, and TF binding sites overlap, illuminating the connection between histone acetylation and gene expression changes. For example, Benaglio and colleagues showed that locus rs6801957 (in the *SCN10A* gene) which is enriched for heart rate, is characterized by high levels of H3K27 acetylation (information from ChIP-seq), high chromatin accessibility (ATAC-seq), and high levels of transcription factor NKX2-5 (ChIP-seq) [155].

**Table 4 cells-11-00200-t004:** Previous epigenetics studies of hiPSC-CMs. Chm indicates chromatin accessibility assay, ac-H indicates histone acetylation assay, gene exp indicates gene expression quantification, and Fxnl indicates functional behavior measurements. “✓” indicates a study’s experimental use of HDAC inhibitors.

Cell Line(s) Used	HDACi Applied	Chm	Ac-H	Gene Exp	Fxnl	Major Findings
In-house-derived hiPSC-CM	✓	–	WB	qPCR	MEA, optical Ca^2+^	TSA improved differentiation towards the cardiac lineage [175].
In-house-derived hiPSC-CM	✓	–	WB	qRT-PCR, microarray	MEA	TSA treatment and suspension culture improve maturity (expression of cardiac genes, homogenous response to hERG blocker) [176].
hiPSC-CM (Axiogenesis)	✓	–	–	microarray	impedance recordings, MEA	HDACi had delayed cardiotoxicity (reduced beat rate, arrhythmic events), HDACi modified pathways related to cell contraction, microtubule/cytoskeleton-based transport, and Z-disc binding [133].
hiPSC-CM (Axiogenesis)	✓	–	–	microarray	–	Panobinostat diminished contractile properties (beat area, beat rate, contraction velocity), increased levels of cardiotoxicity biomarkers (cTnI, FABP3, and NT-proBNP), downregulated cardiac structural and functional genes) [177].
hiPSC-CM (iCell, CDI)	✓	–	–	–	whole-cell patch clamp	Vorinostat reduced I_Na_ current density [178].
26 in-house-derived hiPSC-CM lines		ATAC-seq	ChIP-seq (H3K27ac, NKX2-5)	RNA-seq, WGS	–	NKX2-5 (TF), H3K27ac, and ATAC peaks are associated with enrichment for EKG characteristics such as heart rate, QT interval, QRS duration, and atrial fibrillation. Histone acetylation and TF info from ChIP-seq can be cross-referenced with ATAC-seq peaks and GWAS to illuminate mechanisms of phenotypic effects. dbGaP: phs000924; NCBI: PRJNA285375; GEO: GSE125540, GSE133833 [155].
27 in-house-derived hiPSC-CM lines		Hi-C, ATAC-seq	ChIP-seq (H3K27ac, NKX2-5)	RNA-seq, WGS	–	Contact propensity is a mechanism of regulating gene expression and is positively associated with H3K27 acetylation and gene expression. dbGaP: phs000924 [169].
In-house derived hiPSC-CM		ATAC-seq, DNA methylation	–	RNA-seq	–	Hypoxia and subsequent reoxygenation alter chromatin accessibility (both positively and negatively in various regions), particularly at transcription start sites, indicating the role of hypoxia-induced chromatin reorganization in regulating gene expression. GEO: GSE144426 [174].

Greenwald and colleagues investigated the epigenetic effects of chromatin loop structures through collection of Hi-C data and analysis of previous ChIP-seq, RNA-seq, and WGS datasets [173]. They found a positive association between contact propensity (the probability that loci at loops physically interact) and gene expression as well as with H3K27 acetylation, suggesting chromatin looping as a mechanism of gene expression regulation.

Also of interest are findings of injury-related epigenetics machinery. Ward et al. explored hypoxia/reoxygenation of hiPSC-CMs and observed that hypoxia and subsequent reoxygenation can alter chromatin accessibility both positively and negatively, depending on genomic region [174].

### 5.2. HDACi in hiPSC-CM Differentiation and Maturation

Early studies have investigated HDACi as tools for improving the hiPSC-CM differentiation process or for improving hiPSC-CM maturity after differentiation. HDAC inhibition, such as by TSA, was shown to direct human-induced pluripotent stem cells into the cardiomyocyte lineage [175]. Other techniques for improving cardiac differentiation include treatment with ascorbic acid [179] and electrical or mechanical stimulation [180,181]. Otsuji et al. also showed that TSA treatment during early hiPSC-CM culture improved beat rate and resulted in more homogenous response to hERG blocker E4031, indicating improved maturity [176].

### 5.3. HDACi Cardiotoxicity Testing in hiPSC-CMs

Although many HDACi exist, only a small group are clinically relevant and have been tested on hiPSC-CMs. We describe here the transcriptional and functional observations of these experiments and their applications in HDACi toxicity screening.

#### 5.3.1. Transcriptional Effects of HDACi in hiPSC-CMs

Using hiPSC-CMs, Kopljar and associates studied a suite of HDACi, including entinostat, tubastatin A, vorinostat, panobinostat, and dacinostat [133,177]. Treatment with these HDACi resulted in downregulated cardiac electrophysiology genes (*GJA1*, *GJA5*, *KCNH2*) and z-disc genes (*MURC*, *NEXN*, *RRAD*). Also observed were upregulation of cytoskeleton gene *TUBB2B* and of genes related to heart failure and hypertrophy [133,182,183,184,185,186,187,188]. Panobinostat significantly downregulated cardiac structural genes (*TNNI3*, *FABP3*, *NPPB*, *MYH7*) while increasing levels of cardiotoxicity markers (cTnI, FABP3, and NT-proBN) [177]. Interestingly, transcript levels were altered more severely over time, where dramatic changes in expression of cardiac electrophysiology genes can be seen at 12 h post-treatment (Figure 6A), paralleling clinical reports of delayed cardiotoxicity in human patients.

#### 5.3.2. Functional Effects of HDACi in hiPSC-CMs

In addition to transcriptomics, toxicity can also be assessed through functional measurements. Dacinostat and panobinostat treatment on hiPSC-CMs caused reduced beating rate, reduced contraction amplitude, and even beating arrest [133]. In whole-cell patch clamp experiments, vorinostat significantly reduced I_Na_ current density [178]. Cell-level arrhythmic events such as sustained or prolonged contraction, fibrillation-like pattern, and beating arrest were also observed. Interestingly, researchers noted delayed effects, shown in Figure 6B, where hiPSC-CMs only began to display functional signs of distress 10+ hours after HDACi treatment.

#### 5.3.3. Transcriptional Changes Corroborate Functional Outputs

Transcriptional and functional effects of HDACi on hiPSC-CMs corroborate each other. With downregulation of cardiac electrophysiology and structural genes, impaired functionality is expected, and indeed observed. HDACi treatment caused abnormal beat rates and contraction patterns, which couple with transcriptional abnormalities to demonstrate HDACi cardiotoxicity. Importantly, certain HDACis known to be less toxic (entinostat, tubastatin A) were associated with only mild or no transcriptional or functional effects in vitro. Moreover, transcriptional and functional observations both indicate delayed HDACi cardiotoxicity. Therefore, hiPSC-CMs offer a drug screening model that can recapitulate important aspects of cardiotoxicity and is able to differentiate between highly toxic and mild HDACi.

#### 5.3.4. In Vitro Results Are Consistent with Clinical Observations

Several HDACi, including vorinostat, panobinostat, romidepsin, and entinostat, have been used in the clinic for 5–10 years. Panobinostat and romidepsin are commonly associated with ST/T abnormalities, ventricular tachycardia, and hypertension. Vorinostat and belinostat lead to QTc prolongation and are uniquely associated with severe side effects (grade 3–4), compared to other HDACi, such as entinostat, which are most commonly associated with grade 0–1 effects [130]. These clinical observations of drug severity have been reflected in vitro, where entinostat was associated with mild or no cardiotoxic effects [133].

## 6. Conclusions and Future Outlook

Human-induced pluripotent stem-cell-derived cardiomyocytes are a promising tool for epigenetics studies. hiPSC-CMs can be reliably and sustainably produced and they are adaptable to high-throughput formats. Ability of the cells to provide electrophysiological information is critical for cardiac and cardiotoxicity studies, and previous studies indicate that hiPSC-CMs are capable of detecting drug-induced cardiotoxicity, allowing application in drug screening. Importantly, large-scale and robust epigenetic profiling remains to be carried out for cells that have been optimized towards a more mature phenotype. Moreover, future broader experimentation with pharmaceutical HDAC inhibition will validate the utility of these cells for preclinical investigation of HDACi cardiac therapeutic potential.

## Figures and Tables

**Figure 1 cells-11-00200-f001:**
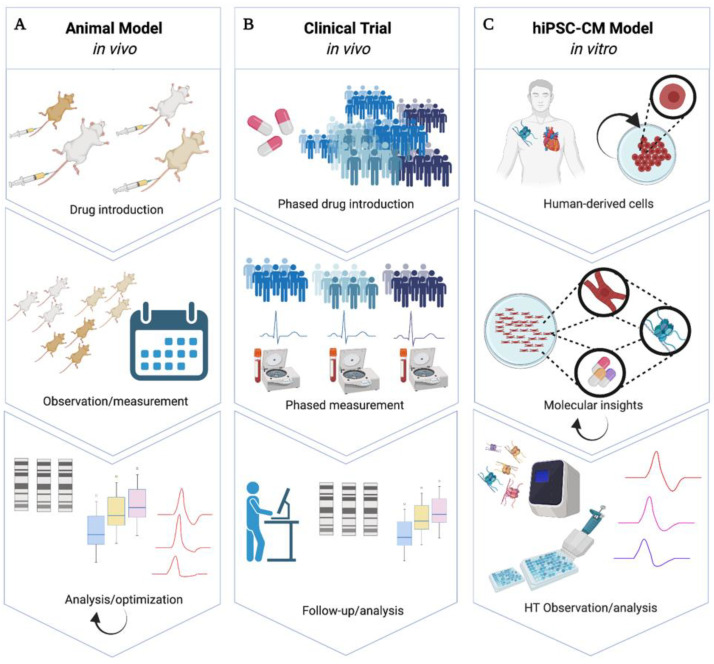
Three main methods of extracting epigenetic information surrounding drug introduction: (**A**) Animal experiments allow chronic observation and biomarker measurement upon drug administration, culminating in postmortem sample collection. (**B**) Multigroup drug introduction in humans is followed by periodic observation and biomarker collection culminating in biomarker analysis and in-person follow-up. (**C**) Patient-derived hiPSC-CMs are used to observe pharmacological cardiac effects in long-term, high-throughput optical and chemical modalities with feedback ability from existing in vitro assays for the direct improvement of future therapy. Created with Biorender.com.

**Figure 2 cells-11-00200-f002:**
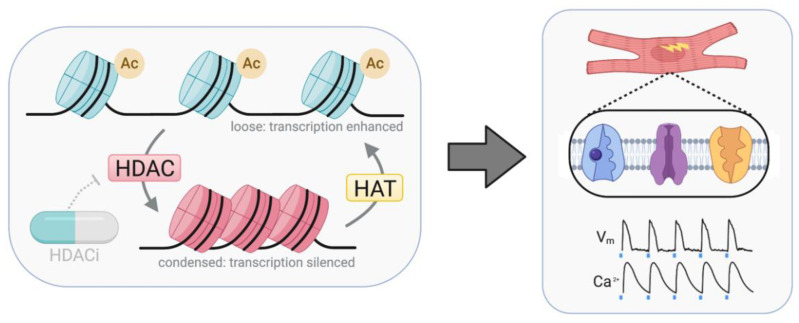
HATs and HDACs epigenetically regulate gene expression through reversible (de)acetylation of histone proteins. Activity of HAT and HDAC enzymes controls chromatin conformation, loosening (HAT) or condensing (HDAC) chromatin structure. Small-molecule HDACi pharmaceuticals disrupt this system and promote transcriptional enhancement. Effects on key cardiac ion channels are observable through functional electrophysiological experiments, e.g., contractility assays or measurements of transmembrane potential (V_m_) and calcium (Ca^2+^) transients. Representative V_m_ and Ca^2+^ transients modified from [15]. Created with Biorender.com.

**Figure 3 cells-11-00200-f003:**
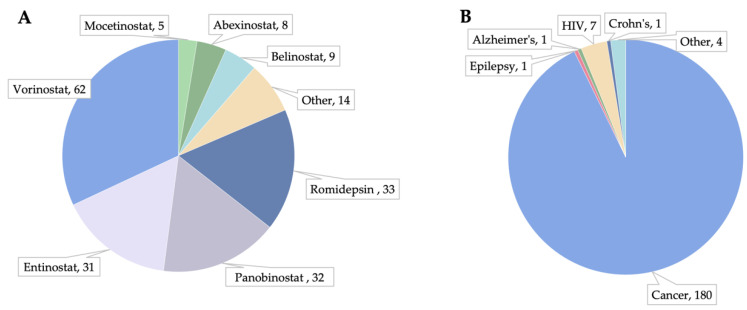
HDAC inhibitor interventions in ongoing US clinical trials (194 total; (**A**)) and their investigated conditions/diseases (**B**), retrieved from clinicaltrials.gov on 22 November 2021. Completed, withdrawn, and/or terminated studies were excluded.

**Figure 4 cells-11-00200-f004:**
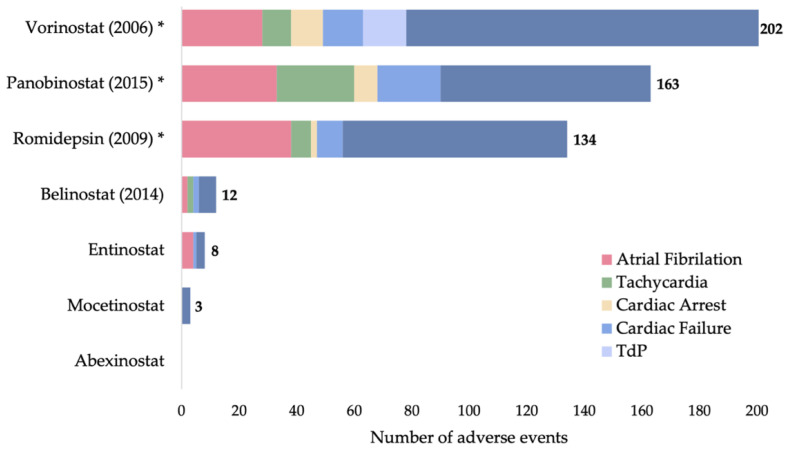
Adverse cardiac cases observed in patients being treated with HDACi (total number of adverse cases reported adjacent to each bar), retrieved from the FDA Adverse Event Reporting System (FAERS) Public Dashboard (fda.gov) on 22 November 2021. FDA approval year, when applicable, is listed in parentheses. * indicates possible TdP risk according to crediblemeds.org [144], accessed on 31 December 2021.

**Figure 5 cells-11-00200-f005:**
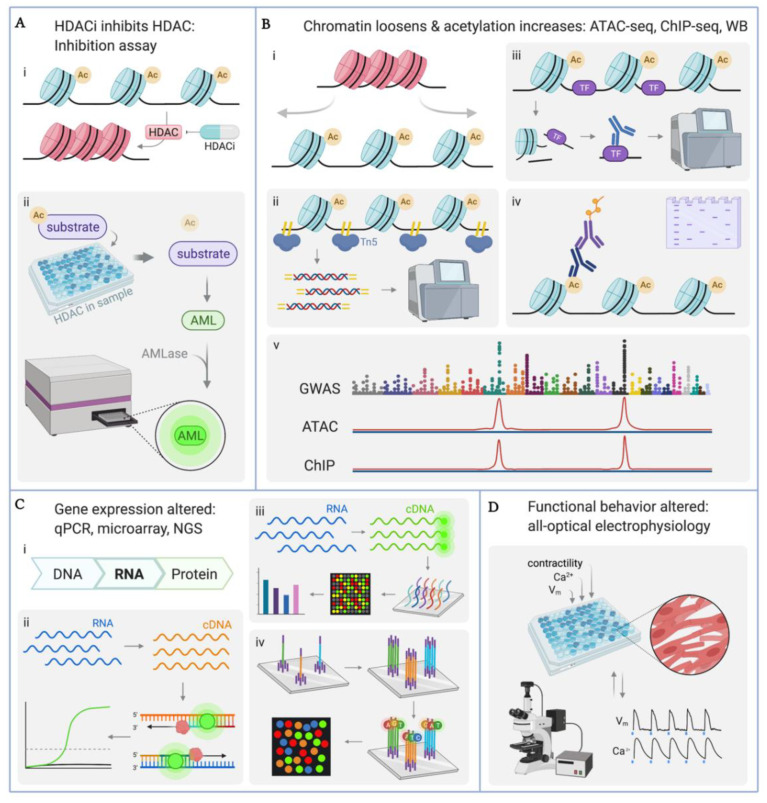
Methods for quantifying effects of HDACi. (**A**) (i) HDACs promote condensed chromatin structure and are counteracted by small-molecule HDACi pharmaceuticals. (ii) HDAC inhibition assay indirectly measures HDAC enzymatic activity. Aminoluciferin (AML), aminoluciferase (AMLase). (**B**) (i) Chromatin shifts from condensed (pink) to loose (blue) structure as histone acetylation increases. Chromatin accessibility can be measured by assay for transposase-accessible chromatin using sequencing (ATAC-seq) (ii) and chromatin immunoprecipitation followed by sequencing ChIP-seq (iii). Transcription factor (TF), Tn5 transposase (Tn5). (iv) Histone acetylation levels assayed with western blot (WB). “Ac” indicates acetylated histones. (v) ATAC-seq and ChIP-seq maps aligned to genome-wide association study (GWAS) maps to reveal areas of open chromatin where TFs of interest and phenotypes of interest are enriched. (**C**) (i) RNA quantification reflects gene expression. (ii) qPCR is a low-throughput fluorescence-based assay. (iii) Microarray assays are a high-throughput alternative to qPCR, allowing simultaneous detection and quantification of thousands of genes. cDNA indicates complementary DNA. (iv) Next-generation sequencing (NGS) is a high-throughput deep sequencing tool. (**D**) Functional behavior is assessed through contractility assays as well as all-optical electrophysiology recordings. Transmembrane potential (V_m_), calcium transients (Ca^2+^). Representative V_m_ and Ca^2+^ transients modified from [15]. Created with Biorender.com.

**Figure 6 cells-11-00200-f006:**
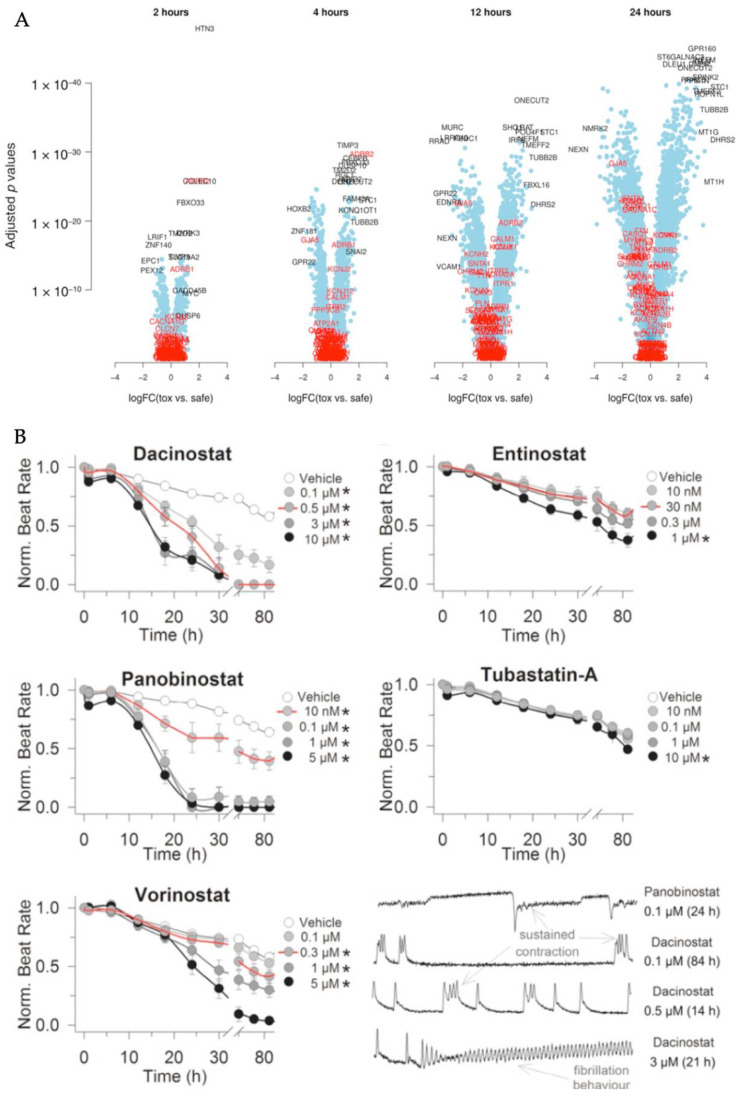
(**A**) Volcano plot illustrates time-dependent transcriptional changes between treatments defined as “toxic” (0.5 mM dacinostat, 0.1 mM and 0.01 mM panobinostat, and 5 mM and 0.2 mM vorinostat) and treatments defined as relatively “safe” (0.3 mM entinostat, 1 mM tubastatin-a, and 0.1 mM vorinostat). Top 10 (based on *p*-value and log ratios, possibly overlapping) differential genes are in black while genes related to cardiac contractility and function are in red. (**B**) Time-dependent changes in beat rate reveal toxic effects of tested HDACi. Red line indicates applicable C_eff_. * *p* < 0.05. Reproduced with permission from [133].

**Table 1 cells-11-00200-t001:** HDACs and their associated roles in cardiac physiology. Chr indicates chromosome location. Heart exp indicates gene expression in the heart. TF indicates transcription factors. KO indicates knockout.

Class	Gene	Chr	Subcellular Localization	Heart Exp	Known Effects on TF	Known Action	Known Cardiac Involvement
I	*HDAC1*	1	nucleus	low	NF-kb, KLF5, YY1, NKX2.5, NR1D2, PER1	H2A, H2B, H3, H4	Promotes cardiogenesis [63]
	*HDAC2*	6	nucleus	high	YY1, KLF4, CRY1	H2A, H2B, H3, H4	Promotes cardiogenesis [63], aids in atherosclerosis models [64], KO increases resistance to hypertrophy [65]
	*HDAC3*	5	nucleus, cytoplasm (shuttles between)	medium	NKX2.5, TBX5, PRARa, YY1, ARNTL/BMAL1-CRY1	H23K27, H3, H4	Promotes cardiomyocyte proliferation [66], KO linked to hypertrophy [67]
	*HDAC8*	X	nucleus (excluded from nucleoli)	medium	TGFb1, RUNX1	H2A, H2B, H3, H4	KO ameliorates pulmonary fibrosis [68]
IIa	*HDAC4*	2	nucleus, cytoplasm (shuttles between)	medium	MEF2, FOXO, TGF-b1	H2A, H2B, H3, H4	KO increases myocardial regeneration, overexpression inhibits cardiomyogenesis [69], inhibition ameliorates I/R injury [70]
	*HDAC5*	17	nucleus, cytoplasm (shuttles between)	low	MEF2, YY1, NKX2.5, PGC-1a, FOXO	H2A, H2B, H3, H4	KO linked to hypertrophy with age [71]
	*HDAC7*	12	nucleus, cytoplasm (shuttles between)	–	MEF2, FOXP3, RARA	H2A, H2B, H3, H4	Promotes hypertrophy [72]
	*HDAC9*	7	nucleus	low	MEF2	H2A, H2B, H3, H4	Suppresses hypertrophy [73], KO attenuates atherosclerosis [74]
IIb	*HDAC6*	X	nucleus, microtubules	low	TGFb1, GATA6	H2A, H2B, H3, H4; misfolded proteins	Promotes fibrosis, KO linked to inhibited fibroblast proliferation [75]
	*HDAC10*	22	nucleus	high	NOTCH1, PAX3, KAP1	–	–
III	*SIRT1*	10	nucleus, mitochondria	low	FOXO, MEF2, HIF1a, PER2, BMAL1	H2A, H3K14, H4K16	Protective against hypertrophy [76], severe overexpression promotes cardiomyopathy [77]
	*SIRT2*	19	plasma membrane, cytoskeleton, nucleus	low	NFAT, FOXO3, HIF1a	H3K56, H4K16	KO increases hypertrophy and fibrosis, decreases ejection fraction [78]
	*SIRT3*	11	mitochondria	high	FOXO, CERS	–	KO promotes hypertrophy and fibrosis [79], KO decreases ejection fraction [80]
	*SIRT4*	12	mitochondria	medium	PPARa	–	Promotes hypertrophy and fibrosis [81]
	*SIRT5*	6	mitochondria, cytoplasm	medium	CPS1, SOD1, SHMT2, CYCS	H3K9	KO promotes hypertrophic cardiomyopathy [82]
	*SIRT6*	19	nucleus	high	NF-kb, HIF1a	H3K9, H3K56	KO promotes hypertrophy [83], protective against I/R injury [84]
	*SIRT7*	17	nucleus	medium	–	H3K18, H3K36	KO promotes hypertrophy and inflammatory cardiomyopathy [85]
IV	*HDAC11*	3	nucleus	–	NOTCH1	H2A, H2B, H3, H4	–

**Table 2 cells-11-00200-t002:** Comparative expression illustrating epigenetic profiles (**A**) * and electrophysiology (**B**) for adult human heart (control) and hiPSC-CM. Blue and red coloring indicates degree of relative over- or underexpression in hiPSC-CMs compared to adult human heart. – denotes genes that were not covered by microarray probe sets. AP indicates action potential. Data acquired from Illumina BaseSpace Correlation Engine, accessed 19 May 2020. * List of cardiac-relevant epigenetics genes derived from [50].

**A**	**Gene Expression Relevant to Cardiac Epigenetics**				
			Gene	Fold change	Ref.
**Writers**	HATs		*p300 (EP300)*	–	
			*pCAF (KAT2B)*	–	
	HMTs		*SMYD1*	–	
			*WHSC1*	1.54	[116]
			*Ezh2*	3.95	[117]
			*SUV39h*	–	
			*DOT1L*	1.42	[116]
**Erasers**	HDAC classes	I	*HDAC1*	1.88	[117]
			*HDAC2*	7.74	[117]
				4.45	[116]
			*HDAC3*	1.51	[116]
			*HDAC8*	1.72	[116]
		IIa	*HDAC4*	–	
			*HDAC5*	−1.28	[117]
			*HDAC7*	1.21	[116]
			*HDAC9*	1.55	[116]
		IIb	*HDAC6*	–	
			*HDAC10*	–	
		III	*SIRT1*	2.24	[116]
				1.79	[117]
			*SIRT2*	−1.96	[117]
			*SIRT3*	1.31	[116]
			*SIRT4*	–	
			*SIRT5*	1.4	[116]
				−4.75	[117]
			*SIRT6*	–	
			*SIRT7*	–	
		IV	*HDAC11*	–	
	HDMs		*Jarid2*	–	
			*Jmjd1*	2.85	[116]
			*Jmjd2*	1.41	[116]
			*Jmjd3*	–	
			*UTX*	2.68	[116]
**Readers**	SWI/SNF		*Brg1 (SMARCA4)*	–	
			*Baf60a (SMARCD1)*	1.99	[117]
			*Baf180 (PBRM1)*	1.43	[117]
				1.25	[116]
			*Baf250 (ARID1A)*	–	
	BETs		*Brd4*	–	
			*14-3-3 (YWHAB)*	–	
	(DDR)-related readers		*ZMYND8 (RACK7/PRKCBP1)*	3.01	[117]
				1.4	[116]
**B**	**Gene Expression Relevant to Cardiac AP**				
Gene			Gene info	Fold change	Ref.
*SCN5A*			Na_V_1.5 → I_Na_	1.48	[116]
*CACNA1C*			Ca_V_1.2 → I_Ca,L_	1.39	[116]
				1.64	[117]
*CACNA1G*			Ca_V_3.1/3.2 → I_Ca,T_	1.41	[116]
*KCNH2*			K_C_11.1 (hERG) → I_kr_	1.51	[116]
				−4.15	[117]
*KCNQ1*			K_V_7.1 → I_ks_	1.51	[116]
				−1.8	[117]
*KCNJ2*			Kir2.1 → I_K1_	−4.24	[117]
*KCNJ12*			Kir2.1 → I_K1_	1.42	[116]
*KCND2*			KV 1.4/1.7/3.4 → I_to,s_	1.39	[116]
*KCND3*			KV 4.2/4.3 → I_to,f_	1.2	[116]
*KCNA4*			KV 1.4/1.7/3.4 → I_to,s_	–	
*KCNA5*			K_V_1.5 → I_Kur_	−1.97	[116]
				−3.37	[117]
*KCNK1*			TWK-1/2 → I_KP_	−2.61	[117]
				−1.92	[117]
*KCNK6*			TWK-1/2 → I_KP_	1.6	[116]
*KCNK3*			TASK-1 → I_KP_	–	
*KCNK4*			TRAAK → I_KP_	1.33	[116]
*KCNJ11*			Kir6.2 → I_K,ATP_	−1.6	[117]
*HCN2*			HCN2/4 → I_f_	–	
*HCN4*			HCN2/4 → I_f_	–	
*ATP1A1*			I_NaK_	2.47	[116]
*ATP1A2*			I_NaK_	−7.15	[116]
				−10.2	[117]
*ATP1A3*			I_NaK_	−1.38	[116]
				−1.6	[117]
*ATP1A4*			I_NaK_	1.46	[116]
*NCX1*			I_NCX_	2.03	[116]
*ATP2A2*			SERCA2	3.59	[116]
				−1.87	[117]
*RYR2*			Ryanodine receptor 2	−4.18	[116]
				−1.87	[116]
*CALM1*			Calmodulin 1	−2.36	[117]
*CALM2*			Calmodulin 2	–	
*CALM3*			Calmodulin 3	–	
*CASQ2*			Calsequestrin	−2.53	[116]
				−80.1	[117]
*KCNIP2*			K^+^ channel interacting protein 2	−1.63	[116]
				−1.44	[117]
*KCNE1*			Auxiliary unit for I_Ks_	−1.73	[117]
*KCNE2*			Auxiliary unit for I_Ks_	–	
*GJA1*			Cx43	–	
*GJC1*			Cx45	1.27	[116]
				1.53	[117]

## Data Availability

See Table 4 for NCBI, dbGaP, and GEO accession numbers. Comparative expression data (Table 2) acquired through Illumina BaseSpace Correlation Engine and is also available through GEO GSE17579, GSE35672.

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
