# Peer review of "Human iPSC-Cardiomyocytes as an Experimental Model to Study Epigenetic Modifiers of Electrophysiology"

_cells, 2022, doi:10.3390/cells11020200_

Round 1
Reviewer 1 Report
The article by Pozo et al., discussed the epigenetic modulators especially, the clinically approved HDAC inhibitors (HDACi) in cardiac toxicity and the discussed the possibility of using iPSC-CMS for drug toxicity screening in vitro.The article included many important details regarding the involvement of HDACs in cardiovascular systems, FDA approved drugs until now etc. However, as a reviewer I believe including the following details will help to connect the title more to the article.
- In Page 7, where you discussed downregulated expression of KCJN2 in iPSC-CMs compared to adult heart and explained possible improvement, adding the article by Li et al.,2017 will be beneficial. They used transduction of KCNJ2 which will be a great method for improvement.
- Page 10 line 244, add citation for each drug.
- Table 3. Add FDA for Romidepsin
- Please include article that reported use of reporter cell lines for cardiac differentiation from iPSCs especially MHHi001-A-5, a GCaMP6f and RedStar dual reporter human iPSC line by Hasse et al.,2021 and TNNT2-luc-T2A-Puro-mCMV-GFP and hACTC-mcherry-WPRE-EF1-Neo Fiedorowicz et al., 2020 which will be very useful for HTS.
- The Figure 6 clarity is too low.
- Advances in electrophysiology measurements, use of chips or usage optogenetic constructs (Optopatch and CaViar) can be discussed.
- A brief discussion about the method to derive cardiomyocytes and disadvantages in detail can be included.
- Briefly discussing some of the advanced techniques in the filed of IPSC-CMs including bioprinting based high-throughput drug testing will be beneficial for readers.
Author Response
We thank this Reviewer for valuable questions and suggestions that helped improve the manuscript. Attached are the point by point responses. Text of the manuscript has been expanded and edited and new references added.

Reviewer 2 Report
The authors in their review entitled "Human iPSC-Cardiomyocytes as an Experimental Model to Study Epigenetic Modifiers of Electrophysiology" make a wide and complete analysis of all the epigenetic mechanisms described. It is well written and very well illustrated. My congrats on this work.
Author Response
Thank you very much for the positive feedback on our review!